# Identification and Validation of a New Peptide Targeting Pancreatic Beta Cells

**DOI:** 10.3390/molecules27072286

**Published:** 2022-03-31

**Authors:** Qianwen Wang, Lei Zheng, Kangze Wu, Bo Zhang

**Affiliations:** 1Department of Surgery, Fourth Affiliated Hospital, School of Medicine, Zhejiang University, Yiwu 322000, China; 22118074@zju.edu.cn; 2Department of Emergency Surgery, the Affiliated Hospital of Qingdao University, Qingdao 266000, China; 11718338@zju.edu.cn; 3Department of Surgery, Second Affiliated Hospital, School of Medicine, Zhejiang University, Hangzhou 310009, China; wukangze@zju.edu.cn

**Keywords:** phage display, targeting peptide, pancreatic beta cells, optical imaging

## Abstract

Noninvasive targeted visualization of pancreatic beta cells or islets is becoming the focus of molecular imaging application in diabetes and islet transplantation studies. In this study, we aimed to produce the beta-cell-targeted peptide for molecular imaging of islet. We used phage display libraries to screen a beta-cell-targeted peptide, LNTPLKS, which was tagged with fluorescein isothiocyanate (FITC). This peptide was validated for targeting beta-cell with in vitro and in vivo studies. Immunocytochemistry (ICC) and fluorescence-activated cell sorting (FACS) analysis were used to validate the target specificity of the peptide. FITC-LNTPLKS displayed much higher fluorescence in beta cells vs. control cells in ICC. This discrimination was consistently observed using primary rodent islet. FACS analysis showed right shift of peak point in beta cells compared to control cells. The specific bind to in situ islet was verified by in vitro experiments using rodent and human pancreatic slices. The peptide also showed high affinity of islet grafts under the renal capsule. In the insulinoma animal model, we could find FITC-LNTPLKS accumulated specifically to the tumor, thus indicating a potential clinical application of molecular imaging of insulinoma. In conclusion, LNTPLKS showed a specific probe for beta-cells, which might be further utilized in targeted imaging/monitoring beta cells and theragnosis for beta-cells-related disease (diabetes, insulinoma, etc.).

## 1. Introduction

Diabetic patients are severely deficient in insulin secretion due to the massive destruction of beta-cell mass (BCM), causing type 1 diabetes [1], or deficient in insulin response due to insulin resistance at early phase and decrease in BCM eventually, causing type 2 diabetes [2]. BCM is a useful indicator of the function of the pancreas. Accurate and non-invasive assessments of dynamic BCM changes during the development of diabetes in humans are critically needed [3]. From another perspective, Islet transplantation has great clinical therapeutic value in type 1 diabetes, and it is also the ultimate treatment for end-stage type 2 diabetes complicated with other organ damage [4]. However, gradual loss of islet grafts mass and function, due to various stress and immune rejections, contributes to the treatment failure [5]. The overtime changes of transplanted islets were largely unknown because of the lack of a clinically validated non-invasive imaging method to track islet grafts.

The discovery of an ideal beta-cell-specific contrast agent is crucial for solving the above problems, which, under certain imaging techniques, could be applied for: 1. quantifying the BCM (beta cell mass) for assessing progression and severity of disease, and timely changing the therapeutic strategies; 2. tracking the function and viability of the transplanted islets over time for detection and intervention of rejection and loss; and 3. theragnosis of beta cell-related diseases, such as insulinoma, especially ectopic, occult tumor.

A series of targets expressed on the membrane of beta cells were considered as beta-cell-restricted marker in previous studies, such as vesicular monoamine transporter 2 (VMAT2), the glucagon like peptide-1 receptor (GLP1R), the dopamine receptor subtype 3 (DRD3), the G-protein-coupled receptor 44 (GPR44), etc. [6,7,8]. By conjugating with a specific contrast agent, the labeled probe targeting the above markers could be utilized to visualize beta cell or islet under certain imaging modalities (MRI, CT, SPECT/PET, optical imaging etc.) [9]. However, due to species-dependent background expression in the acinar pancreas, unfavorable signal-to-noise ratio, and other reasons (size, immunogenicity, pharmacokinetic properties etc.), none of these targets or tracer probes has allowed for successful islet imaging in humans. It is crucial to discover and validate novel beta-cell-specific molecular targets to facilitate imaging and therapy.

Phage display technology is a powerful approach for the generation of peptides and antibodies that target specific organ or tumor structures [10,11,12]. It forms an efficient expression and screening system by combining the ability of the recombinant protein to recognize the antigen. There are promising applications in the field of biotechnology such as monoclonal antibody development and tumor-specific peptides [13]. With respect to islets, S. Ueberberg et al. [14] isolated a novel antibody targeting beta cell by in vitro panning of a phage-displayed library in the rat beta cell line INS-1, and they concluded that in vitro phage display in a beta cell line could be a novel strategy for the isolation of beta cell-specific agents/biomarkers.

Compared to the antibody and its fragments mostly used in the previous study, the screened peptides have several advantages as disease-specific molecular imaging probes. They are made up of a few amino acids and have small structures with molecular weight, leading to high tissue penetration, short plasma half-life, and low associated immunogenicity [15]. In this study, we aimed to develop a novel peptide that specifically binds to pancreatic beta cells/islets through phage display technology screening from beta cell line MIN6, and to further validate its specificity to beta cells. 

## 2. Material and Methods

### 2.1. Cell Lines and Cell Culture

Mice insulinoma cell line MIN6 were obtained from the Academy of Sciences Cell Bank of Shanghai and routinely cultured in RPMI-1640 medium (Gibco, Shanghai, China), supplemented with 10% (*v/v*) fetal bovine serum (FBS) and 1% (*v/v*) penicillin-streptomycin. The murine pancreatic cancer cell line KPC and the hepatocellular carcinoma cell line (HCC-LM3) were kindly provided by Tingbo Liang’s Research Group (Zhejiang University, Hangzhou, China). KPC cells were maintained in RPMI-1640 medium (Gibco, Shanghai, China), supplemented with 20% FBS, 1% sodium pyruvate (100×), and 1% MEM Non-Essential Amino Acids (NEAA 100×). HCC-LM3 cells were maintained in Dulbecco’s modified Eagle’s medium (DMEM; Gibco, Shanghai, China) with 10% FBS and 1% (*v/v*) penicillin-streptomycin. The primary islets were isolated by collagenase digestion from ICR (Institute of Cancer Research) mice and separated by density gradient centrifugation as previously described [16]. The islets were incubated at 37 °C in 2 mL RPMI-1640 medium at a density of 100–150 islets per well in 6-well plates and maintained at 37 °C with 5% CO_2_.

### 2.2. Phage Display Library

The Ph.D.-C7C Phage Display Peptide Library Kit was purchased from New England BioLabs (Beverly, MA, USA) and used to isolate peptide ligands specifically binding to beta cells. This library is constructed based on M13 phage and contains random heptapeptides (with a disulfide constrained loop) fused in-frame to the N-terminus of the minor coat protein (pIII) of M13 phage. The first amino acid of peptide-pIII fusion is the first randomized position. There is a short spacer sequence (Gly-Gly-Gly-Ser) between the displayed peptide and pIII. The library consists of approximately 10^9^ electroporated sequences amplified once to yield approximately 100 copies of each sequence in 10 μL of the supplied phage, which is sufficient to express approximately all of the possible 7-mer peptide sequences that can be expressed from a randomized seven amino acid sequence. 

### 2.3. Biopanning, Selection, Amplification and Sequencing of the Selected Phages

Similar to our previous study [12], a total amount of 2 × 10^11^ pfu mL ^−1^ in 10 μL M13 bacteriophage (Ph.D.-C7C, NEB) were added to MIN6 cells cultured in a tube that was placed on a 3D shaker for 1 h at room temperature. The supernatant was discarded and the cells were washed with 1 mL serum-free RPMI 1640 media (Gibco, Shanghai, China). After four intensive washes using 1 mL PBST solution, the cell-bound phages were eluted by incubation with 600 μL bovine serum albumin (BSA) 0.1 wt% in 0.2 M HCl (pH 2.2) for 8 min at room temperature. One-hundred-and-fifty microliters of Tris-HCl 1 M, (pH 9.0) were added to neutralize the solution and the eluted phage was recovered. Then, the eluted phage was added to the blocked KPC cells, which were placed at 37 °C for 1 h to absorb the phages, which can combine with KPC cells. The remaining phages (not absorbed by KPC cells) were collected for amplification. The *Escherchia coli* (*E. coli)* strain ER2738 as a robust F^+^ strain with a rapid growth rate was exploited for the growth and amplification of a certain phage according to the protocol. The screened and amplified phages were cultured on LB/IPTG/X-gal plates, and the phage titers were determined. The ratio of output to input was calculated (input is phage titer before screen and output is phage titer after screen). The same procedure was repeated another two times. 

At the end of three rounds of screening, the binding peptide libraries of suitable titers were obtained and cultured on LB/IPTG/X-gal plates at 37 °C overnight. Thirty random plaques were selected and added to host *E. coli* ER2738 for amplification. After purification and titering, the phage was prepared for ELISA.

DNA of final selected phages was extracted and sent to Bo Shang Biotechnology (Shanghai, China) for sequencing and then the sequence of our peptide of interest was obtained. 

### 2.4. Enzyme-Linked Immunosorbent Assay (ELISA)

MIN6 cells were seeded in 96-well plates at a concentration of 1 × 10^4^ per well in 200 μL medium/well and placed in a CO_2_ incubator at 37 °C for 24 h. The cells were washed three times with PBS and then fixed with 4% paraformaldehyde for 15 min at room temperature (RT). After fixation, the cells were washed three times with PBS again and blocked with PBS containing 0.05% BSA for 1 at 37 °C. Next, 1 × 10^7^ pfu of candidate phages were incubated separately with MIN6 cells in triplicate at RT for 1 h. The cells were washed with phosphate-buffered saline with Tween 20 (PBST) three times, then 100 μL HRP-labeled anti-M13 mAb (Sinobiological, Beijing, China) were added and cells were incubated at room temperature for 1 h. The mixture was washed with PBST 5 times, and then tetrabenzidine (TMB) (Suo Laibao Technology, Beijing, China) was added to each well, incubating the plate in the dark for 10 min. Sulfuric acid 2 M was used to stop the reaction. The 96-well plates were then measured at 450 nm using an ELISA reader (Bio-Tek ELX 800, Vermont, NE, USA). Irrelevant wild-type phage clone and PBS were used as control groups. All experiments were performed in triplicate.

### 2.5. Peptide Synthesis

The candidate and control peptide were synthesized using standard solid-phase fluorenylmethoxycarbonyl chemistry (Huaan Biotechnology Ltd., Hangzhou, China). Fluorescein isothiocyanate (FITC) was conjugated with N terminal of candidate and control peptide based on kit’s manual instructions from Thermo Scientific. Briefly, FITC was dissolved in DMF at 10 mg/mL, which was further added to the peptide solution for reaction (with 15- to 20-fold molar excess of FITC). The mixture was incubated for 1 h at RT in the dark. Then, the excess and hydrolyzed FITC was removed. The synthesized peptides were characterized by mass spectrometry and high-performance liquid chromatography (HPLC) to confirm their identity and purity. Peptides were greater than 95% pure as determined by HPLC.

### 2.6. Effect of the Selected Peptide on Islet Cell Viability and Insulin Secretion

To investigate the effect of LNTPLKS on the viability of the islet cells, different concentrations of LNTPLKS (0, 0.5, 1, 10 μM) were separately co-incubated with islet cells in 96-well plates (triplicate wells for each concentration of peptide). Twenty-four hours later, MTT was added and incubated for 4 h. DMSO was added after centrifugation and the color development was read at an OD of 570 nm. Islet cells were incubated with different concentrations of peptides for 24 h at 37 °C. The supernatant was discarded, cells were rinsed twice with PBS, KRB buffer containing 3.3 mM glucose was added, and the cells were incubated for 30 min at 37 °C. The supernatant was discarded and cells were incubated with KRB buffer containing low glucose (3.3 mM) or high glucose (16.7 mM) for 1 h. The supernatant was collected and stored at −20 °C for subsequent tests. Insulin assay was performed using an ELISA (Suo Laibao Technology, Beijing, China) according to the kit instructions.

### 2.7. Immunocytochemistry and Flow Cytometry

Purified FITC-labeled candidate peptides were used for the peptide-based immunofluorescence assay to confirm the selective binding of the selected peptides to the used cells. FITC-labeled peptide-based immunofluorescence assay was performed, and FITC-labeled peptide (1 μM) was added to each well and then incubated for 2 h at room temperature. The wells were washed with PBS three times and directly mounted with vectashield mounting medium after DAPI staining. The cells were visualized under fluorescence microscope. In addition, to determine the binding location of the peptide, MIN6 cells were labeled with LNTPLKS-FITC and further observed under confocal microscope. For fluorescence activated cell sorting, purified islets and KPC cells were incubated in 6 well plates; 24 h later, FITC-labeled peptide was added with a final concentration of 1 μM. After 2 h of incubation, cells were washed with PBS three times and harvested for FACS to quantitatively analyze the binding affinity of the peptide to the different cell types (islets was digested to single cells before FACS analyzing).

### 2.8. LNTPLKS-FITC Peptide Binding to In Situ Islets and Islets Grafts In Vitro

In order to evaluate whether the LNTPLKS-FITC peptide could directly bind islets within the pancreas, the pancreas of ICR mice was collected and frozen sections were prepared. The cryosections were incubated with anti-insulin solution and LNTPLKS-FITC 2 h after treatment with BSA, followed by observation under fluorescence microscope. To determine whether LNTPLKS could bind to human islet, staining of paraffin-embedded human tissue sections were performed. Briefly, slices were deparaffinized and subsequently permeabilized by heating in antigen-unmasking solutions. After blocking with 2% BSA, slices were incubated with anti-insulin antibody at 4 °C overnight. Cy3-conjugated goat anti-mouse IgG (1:300 dilutions) antibody was incubated for 50 min at 24 °C. Then, the slices were incubated with DAPI and LNTPLKS-FITC or control peptide for 1 h at room temperature. Tissue slides were analyzed using a fluorescence microscope. To further confirm the specificity of peptide to islet, we transplanted islets under renal capsule of the ICR mice. Kidney slices containing islets grafts were prepared and incubated with LNTPLKS-FITC peptide. After 2 h of incubation, the slices were observed under fluorescence microscope.

### 2.9. In Vivo Biodistribution of LNTPLKS-FITC Peptide

In total, 2 × 10^6^ MIN6 cells and HCC-LM3 cells were subcutaneously inoculated in male nude mice for 4 weeks separately. After 16 days, the tumor grew to approximately 1.0 cm^3^. An amount of 5 mg/kg LNTPLKS-FITC was injected into the tail vein and the mice were sacrificed after 3 h. The tumor, spleen, kidney and liver were harvested and their fluorescence intensity was quantified by in vivo imaging system (IVIS). All animal experiments were approved by the Animal Experimentation Committee of Second Affiliated hospital of Zhejiang University (No. 2015-236; 25 February 2015) and were performed in accordance with relevant guidelines and regulations.

### 2.10. Statistical Analysis

SPSS 16.0 software (IBM, Armonk, NY, USA) was used for all statistical analyses of the present study. Unpaired two-tailed *t*-test or one-way ANOVA was applied to compare quantitative differences between samples in the experiments (mentioned separately for respective figure), and *p* < 0.05 was considered as statistically significant, unless otherwise noted. All data are presented as the mean with SD (standard deviation).

## 3. Results

### 3.1. Isolation of Cell-Targeting Peptide for MIN6 Cells

Three rounds of phage screening were performed on MIN6 and KPC cells. The number of recovered phages increased each round (Figure 1A), and the third round was 90 times more enriched than the first round, indicating that the peptides targeting MIN6 cells are effectively enriched. After the third round of screening, 30 blue plaques were randomly selected and identified by ELISA (Figure 1B). As the result of measuring the absorbance intensity in the 450 nm wavelength, a total of 16 candidate phages were selected (1, 2, 4, 7, 8, 12, 13, 15, 17, 18, 19, 21, 22, 24, 26 and 30), and the OD value was three times or more higher than that of the control group (*p* < 0.05). The selected positive phage clones were then subjected to DNA extraction and sequenced. Amino acid sequence was deduced according to the brief genetic code table provided by the reagent (Figure 1C). The peptide sequence LNTPLKS was with high frequency, implying that an effective enrichment was obtained during the screening process. We also used SPPTGIN-FITC peptide as a negative control.

### 3.2. Evaluation of Specific Binding Affinity of the Peptide-FITC Conjugate

To further evaluate the specificity of the selected peptide, the LNTPLKS and the random peptide SPPTGIN were synthesized and conjugated with FITC at the N-terminus. The purity of the peptide was determined by HPLC, showing a purity greater than 95% (Appendix A). MIN6 cell line was treated with LNTPLKS-FITC or SPPTGIN-FITC, while KPC cell line was treated with LNTPLKS-FITC as a cell control group. A significantly higher FITC signal was observed in MIN6 cells compared to that of KPC cells after labeling with LNTPLKS-FITC. Compared with selective peptide labeling, almost no fluorescence was detected in MIN6 cells labeled with control peptide SPPTGIN (Figure 2A). MIN6 cells belong to the pancreatic beta cell line; it is presumed that the selected peptide might have similar specificity to primary islet. We first examined the toxicity of peptide to islets, which were rather sensitive to various stressors. Therefore, experiments on the effects of the targeting peptide on islet cell viability and insulin secretion were performed. Primary islet cells from ICR mouse were isolated and purified (Figure 3A,B), then treated with a series concentration of selected peptide. The experimental results showed that, when the concentration of the peptide was between 500 and 1 μM, the viability of the islet cells (Figure 3C) and the secretion of insulin (basical insulin secretion, BIS) and glucose-stimulated insulin secretion (GSIS) (Figure 3D) were only slightly affected. Therefore, the concentration of 500 nM and 1 μM were selected in the following studies. To test the affinity of the peptide to islet, purified islets were treated with the selected peptide and control peptide as above. KPC cells were served as cell control. As shown in Figure 3E, fluorescent images showed that the peptides possess high affinity for the islet cells. Flow cytometry demonstrated that the mean fluorescence peak point was shifted to the right side in islet group (and the fluorescent intensity increased along with the concentration of the peptide), while there was no such shift in the KPC cell group, confirming that the LNTPLKS peptide had a higher binding capacity to islet, while no affinity to KPC was found (Figure 3F). To determine the localization of the peptide on MIN6 cells, treated cells were observed under confocal microscopy, demonstrating that the peptide was internalized to the cytoplasm (Figure 2B).

### 3.3. LNTPLKS-FITC Binding to In Situ Islets and Beta-Cell Grafts

The ability of the selected peptide to differentiate islets from the exocrine tissues in pancreatic sections was analyzed. To guarantee that the pancreatic islet cells were identified, anti-insulin immunostaining was carried out as a counterstaining. The random peptide SPPTGIN-FITC was used as control group. As shown in Figure 4A, islets labeled with LNTPLKS-FITC were fluorescent, whereas the surrounding exocrine tissues were not. In the control group, no fluorescent signal was merged with the anti-insulin signal. To further prove the affinity of LNTPLKS to human islets, pancreatic tissue samples from a donor were co-stained by immunohistochemistry using LNTPLKS-FITC peptide and anti-insulin antibody. As shown in Figure 4B, LNTPLKS could be identifiable within the islets and co-localized with insulin-producing cells. Furthermore, we transplanted primary islets (ICR mice) under renal capsule of the ICR mice. The slices containing the grafts were incubated with LNTPLKS-FITC and SPPTGIN-FITC as described in the above procedure. Also these results showed that the islets grafts stained by LNTPLKS-FITC were fluorescent (Figure 4C), while no fluorescence was observed in the control group. These results indicated that the LNTPLKS peptide could specifically bind the islet cells within pancreatic slices and islet grafted transplants under the renal capsule in slices.

### 3.4. LNTPLKS-FITC Targeting MIN6 In Vivo

To evaluate the targeting ability of the selected peptide to MIN6 in vivo, a subcutaneous insulinoma mouse model was established, with HCC LM3 cell tumor as the control model. Mice were sacrificed 3 h after the injection of LNTPLKS-FITC peptide into the tail vein of nude mice. The tumor, liver, spleen, and kidney were removed and the fluorescence intensity was evaluated using IVIS (Figure 4D). Fluorescence average counts showed that the accumulation of fluorescence in the MIN6 tumor (1.80 × 10^3^) was much higher than that of the LM3 tumor (9.47 × 10^2^), suggesting that the peptide had a high affinity for the MIN6 tumor. A distribution of fluorescence was also observed in liver (1.81 × 10^3^, 2.15 × 10^3^) and kidneys (1.14 × 10^3^, 1.20 × 10^3^), indicating that the peptide might be metabolized in these two organs.

## 4. Discussion

In recent years, real-time, noninvasive, and dynamic targeted molecular imaging technologies have played an important role in monitoring the function of islet graft, such as MRI, SPECT/PET, and US [9]. A number of biomarkers such as the sulfonylurea receptor, VMAT2, and GLP1R have been developed for imaging islets in the pancreas [6,7,8]. Recently, the GLP-1 analogue exendin-4 has become a hotspot in the imaging of islet or insulinoma in recent years [17]. Unfortunately, these markers are not truly satisfactory for targeted imaging or further clinical application [18], making it important to find new specific target molecules to image islet grafts in a non-invasive manner.

In this study, the seven-amino acid peptide LNTPLKS that targets the islet beta cell line MIN6 was screened by phage display technology. Short peptides are more suitable for imaging in vivo due to their rapid blood clearance, good biocompatibility, excellent diffusion, and good tissue penetration [15]. After phage enrichment and binding affinity screening, we performed MTT test and glucose stimulated insulin secretion (GSIS) assay to evaluate the potential effect of the selected peptide on beta cell viability and insulin secretion. Then, we validated this peptide not only in vitro using MIN6 cells, islet cells, islet-containing pancreatic slices (both rodent and human), and sections containing islet grafts in islet transplantation model, but also in vivo from animal insulinoma model.

The final goal of developing a beta cell-specific molecular probe is to provide a non-invasive, in vivo imaging method for measuring BCM during diabetes progression and tracking islet grafts after islet transplantation, so that necessary and timely intervention can be performed [19]. Therefore, all presumed beta cell-specific probes screened in vitro have to be validated by in vivo studies. As the first step, we tested islet specificity using pancreatic slices and islet graft-containing slices. The results showed that LNTPLKS-FITC peptide displayed good affinity to islet and could differentiate islet from the background. However, when it comes to in vivo studies, the results were obscure. We injected FITC-conjugated LNTPLKS peptide into the mice via tail vein and the pancreas was then sliced to detect the targeting of the peptide in the pancreatic islet. The fluorescence within pancreas was rather weak (data not shown). The unsatisfied result might be attributed to the degradation of the peptide in the body. Increasing peptide stability and its circulation time will facilitate cellular uptake. In the following study, the peptide LNTPLKS will be further processed by chemical modification (such as PEGylation [20]) to protect the peptide from degradation in vivo without changing the targeting ability of the peptide, or by conjugating it with other imaging materials (iron oxide nanoparticle [17], radionuclide [18], etc.) to be monitored by MR, SPECT, or other imaging modalities with high sensitivity.

A targeted experiment was also performed on insulinoma in vivo, and the results confirmed that the peptide was effectively targeting the insulinoma. Unlike the anatomy and size of the islet in situ and the transplanted cells, the islet tumor cells in the insulinoma are clustered together, which is more favorable to the mass accumulation of the targeted peptide and more easily detected by fluorescence. This indicates the potential clinical application of the peptide on insulinoma detection. The majority of insulinomas are intrapancreatic, scattered, and multiple small-sized (<2 cm) tumors. Moreover, approximately 10–27% of the insulinomas are occult [21]. These aspects result in great difficulties when performing clinical imaging localization. Sowa-Staszczak et al. [22] used 99mTC labeled exendin-4 for the first time to localize small insulinoma tumors in 11 patients, thus it can be also used in our context.

One limitation of the present study is the unknown of the target. In this study, despite the fact that the peptide LNTPLKS showed affinity to islet, the target protein or binding site of this peptide and the mechanism of internalization are not known. Such information is crucial, since any toxicity or antibody formation against the targeting agents during interaction, leading to changes of relative signaling pathway and islet physiology, should be carefully examined before the peptide can be safely applied to humans. Furthermore, it might be much more reliable to reflect beta cell mass and cell function status by in vivo measuring the expression level of the target-protein (via imaging method) and in turn the uptake process. Our next step is to identify the target protein on the islet that the peptide LNTPLKS binds to, and how the peptide interacts with the target and possible effect on islet.

In conclusion, we were able to derive a novel peptide LNTPLKS screened from mouse beta cell line MIN6 based on the phage-display library, and a series of in vitro and in vivo experiments verified its good affinity and specificity to beta cells. Despite the results being quite preliminary, this novel beta cell-targeted peptide offers a promising perspective for non-invasive targeting/imaging of islet in situ, islet grafts, and insulinoma; however, this has to be demonstrated further in future studies. Further experiments are needed to identify the target protein on the islet that the peptide binds to, and to increase the in vivo stability by chemical modification and contrast agent conjugation for improving beta-targeted imaging properties. Also, comparative research should be further conducted between conventional probes (targeting sulfonylurea receptor, VMAT2, GLP1R, etc.) and selected short peptide LNTPLKS, to confirm the latter’s advantages and selectivity.

## Figures and Tables

**Figure 1 molecules-27-02286-f001:**
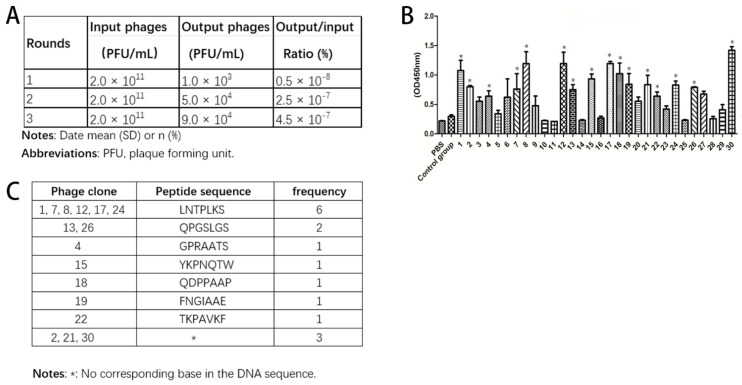
The selection of targeted peptide for MIN6 cells. (**A**). Titer efficacy of the selected phages from each biopanning round. (**B**). Verification of phage adhesion to MIN6 cell through ELISA analysis (triplicate wells for each clone, * *p* < 0.05). Results are mean ± SD, and statistical significance is calculated through unpaired *t*-test. (**C**). The sequence and frequency of positive phage clone.

**Figure 2 molecules-27-02286-f002:**
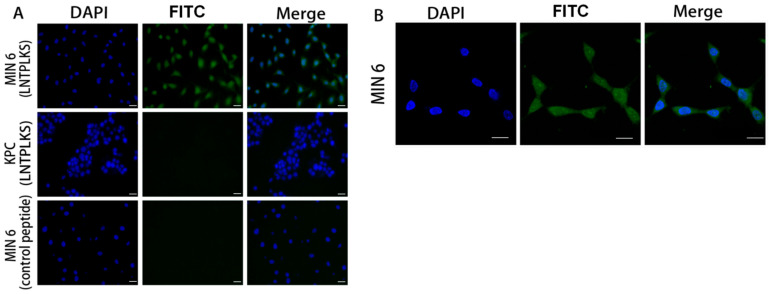
The affinity of targeted peptides for different cells and localization of targeted peptides in MIN6 cells. (**A**). Fluorescent images of MIN6 cells after staining with LNTPLKS-FITC or SPPTGIN-FITC. KPC cells staining with LNTPLKS-FITC were served as cell control. Scale bar = 100 μm. (**B**). MIN6 cells stained with LNTPLKS-FITC and DAPI and observed by confocal microscopy. Scale bar = 100 μm.

**Figure 3 molecules-27-02286-f003:**
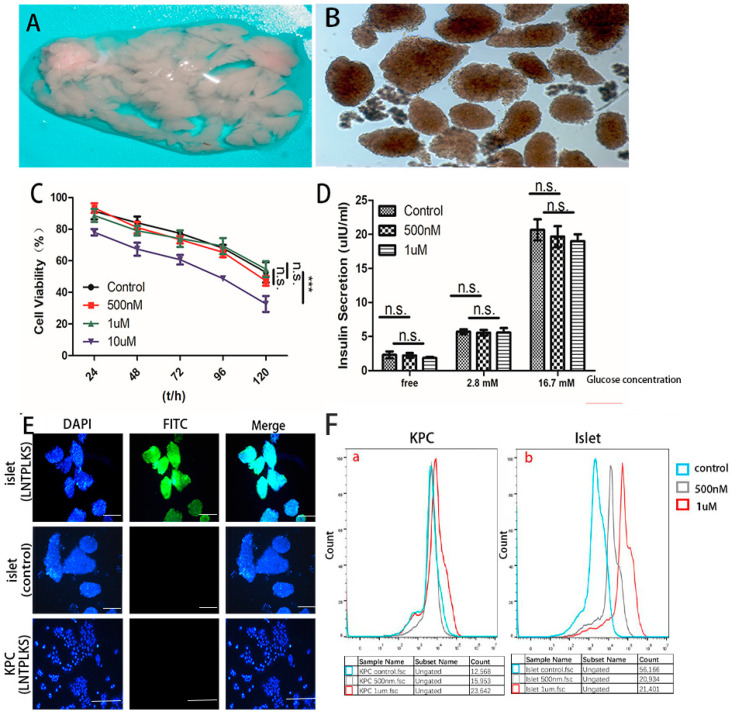
Confirmation of islet cell targeting ability of LNTPLKS-FITC. (**A**). The pancreas of mice was inflated by collagenase. (**B**). Islets were isolated and purified. Image is at 40× magnification. (**C**). The effect of various concentrations of LNTPLKS peptide on islet cells viability (triplicate wells for each concentration of peptide, *** *p* < 0.01, n.s.: not significant). (**D**). The effect of various concentrations of LNTPLKS peptide on insulin secretion stimulated by glucose (three tests were performed. n.s.: not significant). (**E**). Fluorescent images of islet cells after staining with LNTPLKS-FITC or SPPTGIN-FITC. KPC cells staining with LNTPLKS-FITC were served as cell control. Scale bar = 100 μm. (**F**). Assessment of the affinity of the LNTPLKS to islet cells (**b**) and KPC (**a**) by FACS analysis. For (**B**,**C**), results are mean ± SD, and statistical significance is calculated through one-way ANOVA.

**Figure 4 molecules-27-02286-f004:**
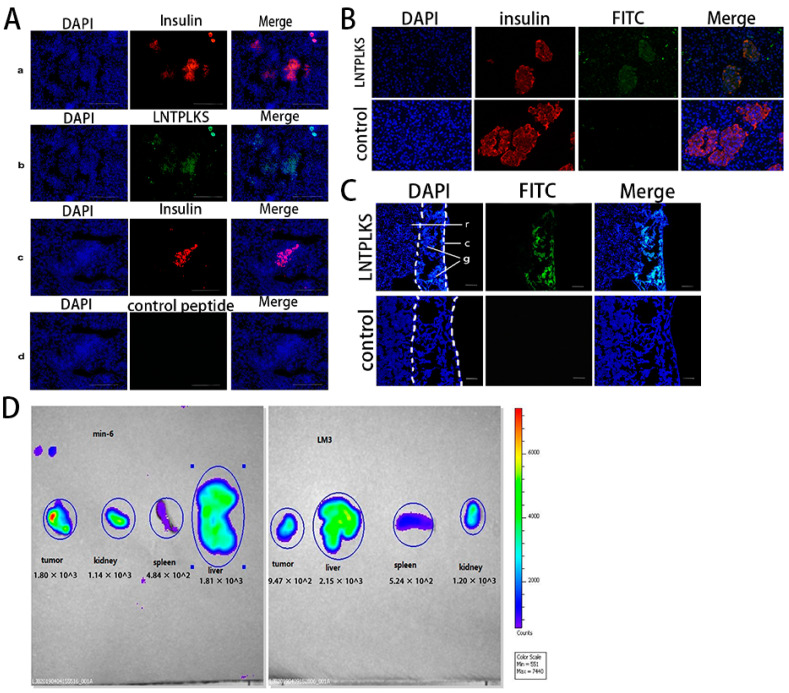
Binding affinity of targeted peptides to in situ islet and islet grafts and bio-distribution of the targeted peptides. (**A**). Immunohistochemical analysis of mice pancreas tissue. Cryosections of pancreas tissue were stained with LNTPLKS-FITC (b) or SPPTGIN-FITC (d) separately. Anti-insulin (a, c) and DAPI stain were used as counterstain for the identification of pancreatic islets in the pancreas tissue. Images are at 40× magnification. (**B**). Immunofluorescence analysis of binding properties of LNTPLKS to human islets. Images are at 40× magnification. (**C**). Fluorescent images of kidney frozen slices (c, capsule of the kidney; g, grafts; r, renal cortex) after staining with LNTPLKS-FITC or SPPTGIN-FITC. Images are at 20× magnification. (**D**)**.** Comparison of bio-distribution of FITC-LNTPLKS in mice bearing insulinoma (**left**) and liver tumor (**right**).

## Data Availability

The data presented in this study are available upon request from the corresponding author.

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
