# Peer review of "Identification and Validation of a New Peptide Targeting Pancreatic Beta Cells"

_molecules, 2022, doi:10.3390/molecules27072286_

Round 1

Reviewer 1 Report

In this article, the authors reported the new peptide, LNTPLKS, targeting pancreatic beta-cell screened by phage display. In vitro and in vivo experiments showed that the LNTPLKS peptide binds to Mice insulinoma cell line Min6 cell and murine pancreatic beta-cell selectively. Furthermore, the authors showed the LNTPLKS peptide shows no side effect on islet cell viability and insulin secretion. The LNTPLKS peptide may become a specific probe for beta-cells.

However, the following points should be considered to overcome several disputes regarding the outcome of the experimental data and the author’s claims. If the following points are improved, the readers will benefit more from this interesting article.

1) Although the authors aimed to develop a specific probe for successful islet imaging in humans, the authors used not a human pancreatic beta-cell line but Min6 mice insulinoma cell line. To prove the LNTPLKS peptide is a species-independent probe, the authors should show that the LNTPLKS peptide also binds to human pancreatic beta-cells selectively.

2) To show the superiority of LNTPLKS peptide from a conventional probe for beta-cells, the authors should use conventional probes targeting beta-cell-restricted markers (i.e., VMAT2, GLP1R, DRD3, and GPR44) and compare the selectivity.

3) Authors should report the HPLC profiles and the mass spectrum of peptides used in this study to show the purity of peptides.

4) Why did the authors use the SPPTGIN peptide as a control peptide? Please explain the reason and design concept.

5) The authors should mention about statistical analysis method for all results.

6) Other careless mistakes were found in various points described below. The authors should prepare the manuscript carefully.

Line 31, 33, 35, 37, 39, 52, 64, 73, 88, 92, 97, 118, 133, 139, 185, 194, 202, 204, 236, 254, 2557, 258, 263, 272, 282, 284, and 293, a space should be inserted in the front of brackets, units, or behind of periods and commas.

Line 65, 66, 266, 268, 270, 273, 276, 303, 308, 309, and 313, “in vitro” or “in vivo” should be shown as an italic letter.

Line 80, 87, 96, 97, 106, and 193, authors should not abbreviate FBS, ICR, PBST, BSA, PBS, BIS, and GSIS due to the first appearance.

Line 127 and 146, “enzyme-linked immunosorbent assay” and “bovine serum albumin” should be abbreviated.

Line 112, 123, and 125, “m/mol” should be shown as “M”.

Line 119, 121, and 140, “hour” should be shown as “h”.

Line 175-177, in Figure 1 legend, the letters “panel A, B, and C” should be shown as bold letters.

Line 192 and 194, “103 nM” should be shown as “1 μM”.

Line 206-207, the title of Figure 2 should not be shown as an italic letter.

Line 233-238, in Figure 4 legend, the letters “panel A, B, and C” should be shown as bold letters.

Line 254, full-width comma should be corrected.

Line 319-334, the authors should correct indents.

Figure 2A and B, “Alexa488” should be corrected to “FITC”

Figure 2A, scale bars should be shown.

Figure 3C, D, and F, a space should be inserted in the front of units.

Figure 3A, B, and E, scale bars should be shown.

Figure 3D, the authors should add the X-axis label meaning Glucose stimulation.

Figure 4A, B, and C, scale bars should be shown.

Figure 4A, merge image should be shown as a merged photo of the three colors (DAPI, anti-insulin antibody, and FITC-peptides).

Reviewer 2 Report

In this original research manuscript the authors describe the discovery of a peptide that target cancerous pancreatic beta cells (Min6 cell line). The authors first panned a phage display library against Min6 pancreatic cells, after three rounds of selection 30 phages were picked randomly and sequenced after amplification, resulting in the identification of epitope sequences. The authors then synthesised one of the peptide and its binding to Min6 was evaluated by ELISA (after conjugation with FITC). The toxicity of the peptide was then evaluated by MTT. The binding of the FITC-conjugated peptide to Min6 was investigated by confocal microscopy, and the ability to use the peptide for sorting cell types was evaluated by FACS. The ability to stain histological slides was then investigated using confocal microscopy. Finally, the peptide was assayed for in vivo staining of tumors by injecting mice that we sub-cutaneously inoculated with cancerous pancreatic cells (Min 6 and HCC-LM3).

The authors claim to have discovered a peptide that selectively target beta-cells, enabling identification of these cells in pancreatic slides. They also claim that the peptide bind to islet grafts as well as in vivo to a pancreatic cell tumor created sub-cutaneously in animal model, providing a clinical application in molecular imaging. The claims of the authors are supported by their data apart for the application for detection of cancer cells as their peptide seems to bind to healthy beta cells as well. The application of targeting the islet for imaging and quantification in diabetes seems already good enough application, why going into cancer without much data to back it up. My major concern is about that a range of details are missing from the description, preventing duplication of the work.

Comment 1: I am extremely surprised that the optimization of peptides to bind to a given cell type without any counter screen against other cell types can result in a selective peptide. How do the authors rationalize their success, was it pure luck? Especially considering that only 30 clones were sequenced, that the enrichment was not huge, and that there is no consensus among the seven sequences that were pulled out.

Comment 2: The method used for chemical synthesis of the peptide is insufficiently described and the information is inappropriately found in the ELISA subsection of the Method section.

Comment 3: How was the library prepared? What is a disulfide constrained library, cyclic etc... Please describe the design of the library.

Comment 4: How were the phages amplified?

Comment 5: How was the candidate peptide synthesized, this is not described? How were the fluorescent probe FITC conjugated to the N-terminus of each peptide?

Comment 6: The authors state that they sequenced 30 phage clones but only 16 are reported in Figure 1C. What happened to the 14 other ones.

Comment 7: In Figure 1C, what does "no corresponding base in the DNA sequence" means? Please clarify.

Comment 8: How was the sequence SPPTGIN used as a negative control? How do you know that this peptide does not bind?

Comment 9: Line 201, replace "specificity" by "affinity". Please check the meaning of "specificity".

Comment 10: Figure 2, the authors state that they use FITC to label their peptide but the figure mentions that they used the other fluorophore Alexa488, what is happening? FITC of Alexa488? In the next figure, Figure 3, then FITC is mentioned. This is confusing.

Comment 11: Figure 2B, please provide information on the statistics used to build the error bars. What are these error bars (SEM?), how many measurements are used for the statistics (n).

Comment 12: Figure 3D, please provide information on the statistics used to build the error bars. What are these error bars (SEM?), how many measurements are used for the statistics (n).

Round 2

Reviewer 1 Report

The authors carefully responded to some issues pointed out. The authors' responses are essential for the readers and the article itself. Revised points enhance the author's claim. If the result of immunohistochemistry analysis to prove the specificity of LNTPLKS peptide to the human islet is added, the revised manuscript will be accepted.

Reviewer 2 Report

The authors have satisfyingly replied all my comments. I think that adding the details of the experiments would increase the impact of your manuscript as it will make it increase the likelihood it could be reproduced.
